# Evaluation of Hygiene Practice for Reducing *Campylobacter* Contamination on Cutting Boards and Risks Associated with Chicken Handling in Kitchen Environment

**DOI:** 10.3390/foods12173245

**Published:** 2023-08-29

**Authors:** Honggang Lai, Yuanyue Tang, Fangzhe Ren, Xin-an Jiao, Jinlin Huang

**Affiliations:** 1School of Tourism and Cuisine, Yangzhou University, Yangzhou 225001, China; lhghstc@163.com; 2Jiangsu Key Lab of Zoonosis, Yangzhou University, Yangzhou 225009, China; tangyy@yzu.edu.cn (Y.T.); fzren@yzu.edu.cn (F.R.); jiao@yzu.edu.cn (X.-a.J.); 3Joint International Research Laboratory of Agriculture and Agri-Product Safety, Ministry of Education of China, Yangzhou University, Yangzhou 225009, China

**Keywords:** cutting boards, hygiene practice, *Campylobacter*, risk exposure

## Abstract

Cutting boards can serve as potential carriers for the cross-contamination of pathogens from chicken to other surfaces. This study aimed to assess chefs’ handling practices of cutting boards across five provinces in China and identify the key factors contributing to unsafe cutting board usage, including cleaning methods and handling practices. Handling practices associated with cutting boards were examined through a web-based survey (N = 154), while kitchen environment tests were conducted to investigate the splashing or survival of *Campylobacter*, inoculated in chicken or on cutting boards, to mimic the practices of chefs. Among chefs in the five provinces of China, wood and plastic cutting boards were the most commonly used for preparing chicken meat. Approximately 33.7% of chefs washed boards with running tap water, 31.17% of chefs washed boards with detergent, and 24.03% of chefs cleaned boards by scraping them with a knife after preparing other meats or chicken. The study tested 23 cutting boards from commercial kitchens for *Campylobacter* presence before and after chicken preparation and cleaning. Among these, 17 were cleaned with a knife, 5 with running tap water, and only 1 with disinfectant. Results showed that cleaning with a knife significantly reduced *Campylobacter* presence on cutting boards (*p* < 0.05), while the three main cleaning methods were inadequate in eliminating contamination to a safe level. In kitchen environment tests, contaminated chicken was chopped on cutting boards, with a maximum distance of 60 cm for low contamination, and 120 cm for medium and high contamination levels. This suggested a contamination risk exposure area ranging from 60 cm to 120 cm. *Campylobacter* survival on surfaces of wood, plastic, and stainless steel was also tested, with plastic surfaces showing the longest survival time (4.5 h at 15 °C and 3.5 h at 25 °C) In comparison, survival time on stainless steel or wood surfaces was only 3 h, implying a cross-contamination risk exposure period of 3 to 4.5 h after chicken preparation. In conclusion, based on the current study data, the practices employed by chefs play an important role in *Campylobacter* transfer in the kitchen environment. The presence of *Campylobacter* on cutting boards even after wiping or droplet splashing highlights its potential as a source of cross-contamination in the kitchen environment. So, chefs in China should reinforce their hygiene culture and adopt effective cutting board cleaning practices to prevent pathogen contamination.

## 1. Introduction

*Campylobacter* spp., which are commonly present in chicken meat, represent a major cause of *Campylobacter* foodborne diseases [1,2,3]. The primary factor contributing to these diseases is cross-contamination between contaminated chicken meat and cooking utensils [4,5]. *Campylobacter* spp. are microaerobic bacteria that pose challenges to cultivation under normal atmospheric conditions [6]. Cross-contamination and inadequate kitchen hygiene practices, such as improper handwashing or inadequate cleaning of surfaces and utensils, play a significant role in *Campylobacter* transmission [7]. Therefore, preventing cross-contamination is crucial. After handling poultry meat, the effectiveness of cleaning may not meet consumer expectations, particularly since microorganisms do not leave visible traces of dirt that can be easily detected [8,9]. Despite efforts to control *Campylobacter*, the global incidence of *Campylobacter*-related illness has not shown a significant decrease over time [10].

*Campylobacter* can easily spread from contaminated chicken meat to various kitchen equipment, including cutting boards, clothing, and knives [11]. Different materials, such as wood, plastic, or stainless steel, are commonly used for cutting boards in domestic and food service kitchens. However, wooden cutting boards have been found to harbor microbial contaminants and pose challenges in terms of effective cleaning [12]. Some studies demonstrated that cutting boards played a role in cross-contamination of foodborne pathogens from poultry meat to cucumber [8,13]. The porous nature of wooden surfaces allows bacteria to penetrate easily, increasing the risk of cross-contamination incidents [12]. Furthermore, in China, the national food safety standard GB 14,934 requires that foodborne pathogens should not be detectable on food contact surfaces such as cutleries, boards, and knives [14].

In China, it has been reported that 77.41% of chicken carcass samples and 37.37% of kitchen surfaces show *Campylobacter* spp. contamination in commercial kitchens [11]. This study aimed to address the knowledge gap concerning the variability in chefs’ behavioral preferences across different provinces, as well as the spread and survival potential of *Campylobacter* and cross-contamination practices during the preparation of raw chicken on cutting boards in the kitchen environment, and study the effectiveness of different hygiene procedures for reducing *Campylobacter* contamination on boards and risks while chicken handling in the kitchen environment. This work was conducted as part of a comprehensive research project and ran parallel to ongoing baseline microbiological surveys on chicken handling in various provinces of China.

## 2. Materials and Methods

### 2.1. Web-Based Survey among Chefs in Eastern China

A web-based survey was conducted to investigate hygiene and handling practices for chicken preparation in commercial kitchens across different regions of China (Jiangsu Province. Guangdong Province, Fujian Province, Guizhou Province, and Hunan Province) with the aim of capturing geographical and cultural variations. The study protocols were reviewed and approved by the Yangzhou University Human Research Ethics Committee (Permit No. YZUHL20210114). The study included chefs over the age of 18 who were responsible for chicken handling in commercial kitchens. Participants were recruited through the Internet for the study in five provinces of China. The aims and objectives of the study were explained to each participant, and the confidentiality of their information was verbally confirmed. The recruited participants were given the choice to complete the questionnaire using Wenjuan Star, an online platform, with options for immediate or later completion. Data collection took place between July and August 2021.

The questionnaire consisted of 12 multiple-choice questions specifically designed to gather information about self-reported practices related to the last instance when chefs prepared raw poultry in a commercial kitchen. The use of a multiple-choice approach aimed to minimize subjectivity and ensure clarity in measuring and addressing various aspects of the topic. A total of 154 completed questionnaires were collected from chefs located in five provinces of China, as presented in Table 1. The majority of respondents in the kitchens were male. The age group of 18–35 was well-represented in all five provinces. Education levels varied among the respondents, and notably, the proportion of individuals with tertiary qualifications was significantly higher in Guizhou and Hunan provinces compared to the other provinces.

### 2.2. Sample Collection and Campylobacter Examination Method

In Jiangsu province, China, a total of 23 commercial kitchens were visited for sampling. During the sampling process, the specialized chefs were instructed to handle chicken carcasses following their regular daily preparation routines in the designated area for raw meat preparation. Prior to chicken preparation, two sterilized cotton balls soaked in physiological saline solution were used to wipe 250 cm^2^ of the exterior surface and 250 cm^2^ of the interior surface of the chicken carcasses. Samples from cutting boards were collected at three specific stages: before handling the chicken, after handling the chicken, and during the cleaning procedure. For cutting boards, two sterilized cotton balls soaked in physiological saline solution were used to wipe a surface area of approximately 100 cm^2^, as described in a previous study [11,15].

The enumeration and isolation of *Campylobacter* spp. were conducted using the plating method. The original solution was diluted 10-fold in a PBS solution, and 100 μL of each dilution was spread onto *Campylobacter-*selective agar base (modified CCDA; Oxoid, UK) plates supplemented with selective antibiotics. The plates were then incubated under microaerobic conditions (5% O_2_, 10% CO_2_, and 85% N_2_) at 42 °C for 36–48 h. Each dilution was plated in duplicate to ensure accuracy. All colonies exhibiting the characteristic morphology of *Campylobacter* were counted within the countable dilution range of 15–300 colony-forming units (CFU) per plate. For further identification, up to five *Campylobacter* colonies were selected from the plates. For species identification, multiplex PCR was conducted using specific primers as recorded in a previous study [11]. The PCR amplification targeted the 16S rRNA gene for all *Campylobacter* species, the mapA gene for *Campylobacter jejuni*, and the ceuE gene for *Campylobacter coli*.

### 2.3. Simulating the Handling of Contaminated Chicken in the Kitchen Environment

To reduce bacterial loads on the chicken samples, the samples underwent irradiation at the Yangzhou Gamma-Ray Center using a dosage of 25 kGy. After irradiation, the samples were carefully placed in an ice box and transported to the laboratory for future utilization. In the laboratory, each bag containing the whole chicken was immersed in a 500 mL bag containing the *Campylobacter* ATCC12662 cultures (10^6^, 10^7^, 10^8^ log_10_CFU/mL). The bags were then shaken for 30 min at 120 rpm to ensure even distribution of the bacteria on the surface of the whole chicken samples. After the incubation period, the whole chicken samples were removed from the bags and transferred to a clean bag for a duration of 2 min. Subsequently, the samples were immediately placed into a sterile bag, sealed, and stored at a temperature of 4 °C for later use. The chickens were then transported to a model kitchen at Yangzhou University, where they were chopped into smaller pieces by a group of volunteers consisting of culinary students attending Yangzhou University. The study protocols were reviewed and approved by the Yangzhou University Human Research Ethics Committee (Permit No. YZUHL20210146). Following the handling of the chicken by volunteers, the sample sites indicated in Figure 1 were wiped using two sterilized cotton balls soaked in physiological saline solution. Cotton balls were utilized to wipe a surface area of approximately 100 cm^2^ in order to collect any potential bacterial contaminants.

### 2.4. Simulating the Handling of Contaminated Chicken in a Kitchen Environment

Chicken juice, also referred to as meat exudate, was prepared following the methodology described in a previous study [16]. To initiate the preparation process, frozen whole chickens were procured from various supermarkets located in Yangzhou City, China. The frozen whole chickens were allowed to thaw overnight at room temperature. Subsequently, the exudate, or the juice released from the meat, was collected. To remove any debris, the collected exudate was subjected to centrifugation. To ensure sterility, a sterile polyether sulfone syringe filter with a pore size of 0.2 μm (Millipore) was used for filtration. The resulting chicken juice was divided into smaller aliquots and stored at a temperature of −20 °C until it was ready for use in subsequent experiments or analyses.

First, the cultured standard strain ATCC12662 broth was diluted separately with PBS (phosphate-buffered saline) and chicken juice. The dilution was then adjusted to achieve an optical density (OD) value of 0.05, corresponding to a concentration of approximately 10^7^ CFU/mL. Next, 500 μL of the bacterial suspension was inoculated onto the surface of different materials, such as stainless steel, plastic, or wood. Each material had dimensions of 5 cm × 5 cm. After inoculation, the material samples were on a shaking table at temperatures of 25 °C and 15 °C, respectively. A time interval of 0.5 h was allowed between measurements. At each designated time interval, three pieces of the inoculated materials were removed and the number of *Campylobacter* on their surfaces was measured. This was achieved by diluting the bacterial suspension obtained from each piece and performing a plate count method.

### 2.5. Calculations and Statistics

The information obtained from the completed questionnaires was recorded and managed using Microsoft Excel 2010 for efficient data handling and organization. Subsequently, descriptive data analysis was performed using Stata 11.0 software. This involved calculating frequencies and percentages of responses within each category. The results of this analysis were presented in tabular form, providing a clear and concise summary of the data, which allows for easy interpretation and understanding of the distribution of responses across different categories. The chi-square test, a statistical test commonly employed to assess the association between categorical variables, was utilized in the analysis.

The data analysis was performed using IBM SPSS v.21 software. Loads between samples were compared using the independent sample *t*-test. A level of significance of 0.05 was applied for all statistical comparisons.

## 3. Results

### 3.1. Chefs’ Self-Reported Handling Practices While Preparing Chicken

In the web survey, a total of 154 chefs who used cutting boards for chicken preparation were consulted (Table 2). The survey aimed to gather information about the types of cutting boards used, the methods of washing hands and knives, and overall hygiene practices. Among the surveyed chefs, similar proportions reported using plastic boards (50%) and wood boards (44.16%). A small percentage of respondents (1.95%) used stainless steel boards, while 3.90% reported using other types of boards. Differences in cutting board preferences were observed across different provinces in China. In Jiangsu Province, wood boards were the most common choice, with 55.56% of respondents using them. In Guangdong Province, plastic boards were the most prevalent, with 62.50% of respondents using them. In Fujian Province, 6.06% of chefs used stainless steel boards, while in Jiangsu province, the usage was 2.78%. Regarding other materials used for cutting boards, 4.17% of respondents in Guangdong Province, 10% in Hunan Province, and 11.76% in Fujian Province reported using them. These findings highlighted the variations in cutting board preferences among chefs in different provinces, with wood and plastic being the most commonly used materials. The data are summarized in Table 2, which provides a comprehensive overview of the distribution of cutting board types across the surveyed provinces.

It was found that the cutting boards were often used for cutting other types of meat in advance of being used to prepare chicken meat. Among the surveyed chefs, the highest percentage of prior use of boards for cutting other types of meat was reported in Hunan Province (55%), followed by Fujian Province (36.36%). When it came to cleaning the boards after cutting other meats, different methods were reported. A total of 33.12% of chefs washed the boards with running tap water, 31.17% scrapped the boards with a knife, and 35.71% washed the boards with detergent. In Guangdong province, the preferred method was washing with running tap water (45.83%), followed by scrapping with a knife in Fujian (42.42%), Guizhou (41.18%), and Hunan province (40.00%). In Jiangsu (47.22%) and Guizhou province (47.06%), a higher percentage of chefs reported the use of detergent for cleaning the boards. After the chicken preparation process, 48% of chefs reported using cloths to clean chicken juice off the kitchen surface. Regarding the cleaning of cutting boards, after chicken preparation, 41.67%, 39.39%, and 33.77% of chefs in Guangdong, Fujian, and Jiangsu province, respectively, cleaned the boards with running tap water. In Hunan and Guizhou Provinces, 45% and 41.18% of chefs, respectively, kept the boards clean by scrapping them with a knife. It was worth noting that the majority of chefs (54.55%) expressed a preference for washing cutting boards with disinfectant. Additionally, half of the chefs reported choosing to wash their knives with disinfectant, which was considered a safe handling practice. These findings shed light on the cleaning practices and preferences of chefs regarding cutting boards and knives after handling different types of meats. Proper cleaning and disinfection methods were crucial to ensuring food safety and preventing cross-contamination in the kitchen.

### 3.2. Campylobacter Contamination of Cutting Boards in Commercial Kitchens during Chicken Preparation

Table 3 presents the different methods used to clean cutting boards during chicken preparation. In 17 kitchens, chefs cleaned their boards with knives. Prior to chicken preparation, boards from 4 out of the 17 kitchens tested positive for *Campylobacter*. However, after chicken preparation, positive tests for *Campylobacter* on the boards significantly increased to 94.12%. Despite following cleaning procedures, boards from 9 out of 17 kitchens still tested positive for *Campylobacter*. Fortunately, the average *Campylobacter* loads on the boards noticeably decreased from 2.97 ± 0.94 Log_10_CFU/100 cm^2^ to 2.10 ± 0.56 Log_10_CFU/100 cm^2^ (*p* < 0.05). In five kitchens, no *Campylobacter* was detected on any of their boards prior to cutting chicken. However, after chicken preparation, all of the boards tested positive for *Campylobacter*, with an average load of 3.44 ± 0.85 Log_10_CFU/100 cm^2^. When the boards were cleaned with running tap water, three out of the five boards were still contaminated with *Campylobacter*, with an average load of 2.85 ± 0.47 Log_10_CFU/100 cm^2^. Only one kitchen used disinfectant to clean its boards. Before cutting, *Campylobacter* was not detected on the boards of that kitchen. Unfortunately, after chicken preparation, the boards were contaminated with *Campylobacter* at a load of 3.76 ± 0.00 Log_10_CFU/100 cm^2^. Even after cleaning with disinfectant, the boards still tested positive for *Campylobacter*, with an average load of 1.6 ± 0.00 Log_10_CFU/100 cm^2^.

### 3.3. Risk Area of Campylobacter Spread during Simulated Chicken Preparation

Figure 2 displayed the variation in *C. jejuni* levels in chicken carcasses that were initially contaminated with similar concentrations of ATCC12662 (10^7^ CFU/100 cm^2^ high; 10^6^ CFU/100 cm^2^ medium; or 10^5^ CFU/100 cm^2^ low). This difference in contamination levels persisted across all three groups, indicating that the initial contamination level had a consistent impact on the subsequent *C. jejuni* levels.

As Figure 3a,b show, a significant negative correlation between the distance and *Campylobacter* contamination on sample sites in the kitchen was found. The results indicated that as the distance increased, the contamination of *Campylobacter* on the sample sites decreased. Notably, in a model kitchen environment, where the maximum detection distance for the low pollution group was 60 cm and for the medium and high pollution groups was 120 cm, there was a similar average number of positive sample sites. However, when the distance reached 60 cm, the reduction in *Campylobacter* loads was not statistically significant (*p* > 0.05).

### 3.4. Risk Time of Campylobacter Survival on Different Surfaces

To simulate the survival of *Campylobacter* on boards made of different materials during chicken preparation in a kitchen environment, *Campylobacter* was added to a suspension containing chicken juice and PBS.

The presence of chicken juice significantly increased the survival time of *Campylobacter* on various abiotic surfaces at both 15 °C and 25 °C. As Figure 4 demonstrates, there was a clear trend of a longer survival time for *Campylobacter* in the presence of chicken juice compared to PBS. In all cases involving chicken juice, the longest survival time was observed on plastic surfaces, reaching up to 4.5 h at 15 °C (Figure 4b) and 3.5 h at 25 °C (Figure 4d). On stainless steel and wood surfaces, the survival time was shorter, with *Campylobacter* surviving for only 3 h. Notably, *Campylobacter* exhibited a faster decline in survival on wood surfaces compared to plastic and stainless steel surfaces at 15 °C.

## 4. Discussion

In the present study, we conducted a survey among chefs from commercial kitchens across five provinces in China to gather information on their routine practices regarding the usage and hygiene of cutting boards. We then evaluated different cleaning procedures to assess their effectiveness in eliminating *Campylobacter* contamination on the boards. Additionally, in laboratory experiments, we simulated chicken preparation on cutting boards and tested the survival of *Campylobacter*. In this section, we will discuss the findings in relation to the risks associated with chicken preparation based on our experiments.

In the UK, it was found that 75% of men and 17% of women do not consistently wash their hands after raw food preparation; similarly, in the United States, data show that 20% of people do not wash their hands with soap [17]. In Egypt and Iraq, alarming statistics revealed that 90% of consumers do not wash their hands promptly after handling chicken. This lack of hand hygiene after poultry handling posed a significant risk of the spread of bacteria and foodborne illnesses [18]. In our study, we found that 41.56% of chefs reported washing their hands with running tap water, while 54.55% of chefs stated that they wash their hands with disinfectant. These findings highlighted different handwashing practices among chefs in the study population. Eriksson’s hypothesis suggested that hands play a crucial role in the transmission of *Campylobacter*. According to this hypothesis, thorough handwashing using antibacterial substances is considered critically important in preventing the spread of *Campylobacter* [5]. However, it is unfortunate that not everyone in commercial kitchens in China recognizes the importance of proper hand hygiene. Adequate hand hygiene practices, including thorough handwashing with antibacterial substances, are crucial in food preparation settings to prevent the spread of bacteria and foodborne illnesses. It is encouraging to note that a significant majority of consumers in New Zealand, specifically 97%, expressed their intention to use separate knives and cutting boards when preparing chicken compared to other food materials [19]. It is concerning to note that in two Middle Eastern countries, a significant portion of respondents, specifically 28.8% and 6.5%, respectively, expressed their unwillingness to use separate cutting boards when handling chicken [20]. Our study revealed concerning findings regarding the practices of chefs in five provinces in China in relation to cutting boards used for chicken preparation. Specifically, 40.91% of chefs reported using the same cutting boards for cutting other types of meat before preparing chicken. Additionally, only 35.71% of chefs chose to wash the cutting boards with detergent (Table 2). In a study conducted in Norway, it was found that 8% of respondents reported using a cloth as their preferred utensil for cleaning [21]. A significant percentage of chefs in five provinces in China (ranging from 41.18% to 61.11%) expressed a preference for using cloths to clean cutting boards during chicken preparation. The presence of *Campylobacter* has been detected in cloths following the preparation of raw poultry [22,23]. Kitchen cloths can potentially serve as vehicles for cross-contamination of pathogens, transferring them from food spills to other foods or food contact surfaces. Indeed, the education of chefs and kitchen staff is crucial in ensuring food safety in commercial kitchens. The study conducted by Mihalache [24] highlighted the importance of making chefs aware of the critical moments when they need to clean their hands, utensils, and surfaces.

Cutting boards made from a variety of materials, including wood and plastic, are commonly used in wet markets worldwide. However, wood cutting boards have gained attention as potential hazardous surfaces only since 1990 [12]. Research has shown that wood cutting boards can harbor microbial contaminants and present challenges in terms of effective cleaning. In the current study, plastic and wood cutting boards were prevalent in commercial kitchens. It is important to note that wooden surfaces are porous and can facilitate bacterial penetration, thereby increasing the risk of cross-contamination incidents [25]. The study also revealed that cutting boards retained significant levels of *Campylobacter* even after the initial wiping. In fact, the amount of *Campylobacter* detected in the second wiping surpassed the limit of detection (100 CFU/mL) for all samples contaminated with ST-918 and for 18 out of 20 samples contaminated with ST-257 [5]. This finding suggests that traditional cleaning methods may not effectively eliminate *Campylobacter* from cutting boards. Furthermore, the use of disinfectants for cleaning cutting boards has been debated. Some studies have indicated that the use of disinfectants can potentially contribute to the emergence of disinfectant-resistant bacteria and may not completely eradicate antibiotic-resistant bacteria [26]. In our study, despite chefs choosing to clean the boards with disinfectants in one commercial kitchen, *Campylobacter* was still detected. This finding can be attributed to the formation of biofilms by foodborne pathogens on food contact surfaces [27]. In a study conducted in France, the adhesion ability of selected *Campylobacter* isolates to inert surfaces was investigated to explore its association with their transferability. It was found that all the characterized isolates from chicken skin samples demonstrated adhesion to inert surfaces, with more than 90% (25/27) of the isolates exhibiting a moderate to high adhesion ability [28]. So, we should reconsider and revise the strategy for controlling bacterial contamination on cutting boards in kitchen settings. One potential approach, the use of composite materials to cover cutting boards and other polymeric surfaces in meat processing environments, could inhibit the growth of foodborne pathogens, and should be recommended [29].

A study emphasized that contaminated water from the process of washing raw poultry had the potential to travel a significant distance, reaching up to 28 inches (71 cm) on both sides of the sink and 20 inches (51 cm) in front of the sink. Disturbingly, a portion of the chefs in our study (12.99%) reported not eviscerating the whole chicken, and a majority (91.56%) preferred washing the chicken before cutting, both of which could be considered unsafe practices leading to droplet splashing containing *Campylobacter*. During the simulation of chicken preparation in the kitchen, it was observed that *Campylobacter* could be splashed as far as 60–120 cm away from the cutting boards, indicating a high-risk zone extending up to 120 cm where other materials could be exposed to a high risk of contamination (Figure 3). In a previous study, it was reported that sinks and their immediate surroundings (within 0–15 cm) exhibited the highest frequency of contamination following the preparation of chicken thighs inoculated with E. coli DH5-α [30]. Similarly, our study demonstrates that cutting boards and their nearest vicinity (0–30 cm) have the highest frequency of *Campylobacter* contamination after chicken chopping. It is widely acknowledged that cross-contamination is the primary route of *Campylobacter* transmission in the kitchen [8,11,31,32,33,34]. However, the findings suggest that droplet splashing can also be an important route for the spread of *Campylobacter* in the Chinese kitchen environment, requiring increased attention. Therefore, it is crucial to design and equip kitchen layouts that effectively prevent the spread of bacteria, thereby interrupting potential transmission pathways.

*Campylobacter* does not survive for extended periods on food contact surfaces such as equipment, countertops, cutting boards, or kitchen utensils [35]. However, in the presence of wet and cold refrigeration conditions, *Campylobacter* could survive on dry surfaces for several days. In our study, *Campylobacter* was found to survive on different materials used to make cutting boards for a risk exposure time of 3 to 4.5 h when exposed to chicken juice, which can enhance the probability of transmission from boards to other materials (Figure 4). The presence of other co-contaminants has been suggested to enhance the survival of *Campylobacter* in adverse environmental conditions [36]. Some common bacterial species associated with poultry-contaminated boards, such as *Aeromonas* spp., *Brochothrix campestris*, *Enterobacter cloacae*, *Pseudomonas putida*, *Serratia marcescens*, *Staphylococcus aureus*, and *Streptococcus* spp., have been identified as emerging pathogens or food spoilage organisms [12]. *S. aureus* has been shown to enhance the survival of *Campylobacter* strains under adverse conditions, including low temperatures and aerobic environments [37]. It is important to consider the potential presence of a microbial community on cutting boards, which may contribute to the longer survival of bacteria in the kitchen environment compared to in laboratory tests. The findings of our study suggest that the three cleaning methods evaluated are not effective in completely eliminating *Campylobacter* contamination on cutting boards. Therefore, the effectiveness of cleaning measures and the issue of bacterial disinfectant-resistance must be carefully considered. Additionally, attention shall be given to the hygiene of knives, hands, and cloths to avoid secondary contamination of cutting boards and food materials.

## 5. Conclusions

The objective of this study was to assess the usage and hygiene practices associated with cutting boards among chefs in five provinces of China. The study aimed to evaluate the impact of different cleaning methods and identify the areas and time intervals after chicken preparation that pose a risk for foodborne illnesses. The findings indicated that wood and plastic cutting boards are widely used in commercial kitchens across the five provinces in China studied, and commonly employed when cutting various materials, including chicken. The study also examined the different cleaning methods employed by chefs, revealing that 41.56% of chefs chose to wash their cutting boards with detergent. Among the provinces, the occurrence of this cleaning practice was higher in Jiangsu Province (41.67%), Guangdong Province (39.58%), Fujian Province (42.42%), and Hunan Province (52.94%). However, even with this cleaning method, there still may be a risk of *Campylobacter* spreading in their kitchens. The paper highlighted the potential sources of cross-contamination in the kitchen environment, such as droplet splashing of *Campylobacter* or the survival of the pathogen on cutting boards even after cleaning. These factors contributed to the risk of *Campylobacter* transmission. In conclusion, it is recommended that chefs in China should reconsider their strategies for effectively cleaning cutting boards to prevent pathogen contamination. Specifically, attention should be given to the risk of *Campylobacter* splashing and surviving on cutting boards. By addressing these concerns, the probability of *Campylobacter* spreading to humans can be significantly reduced, thereby mitigating the risk of foodborne illnesses in commercial kitchens.

## Figures and Tables

**Figure 1 foods-12-03245-f001:**
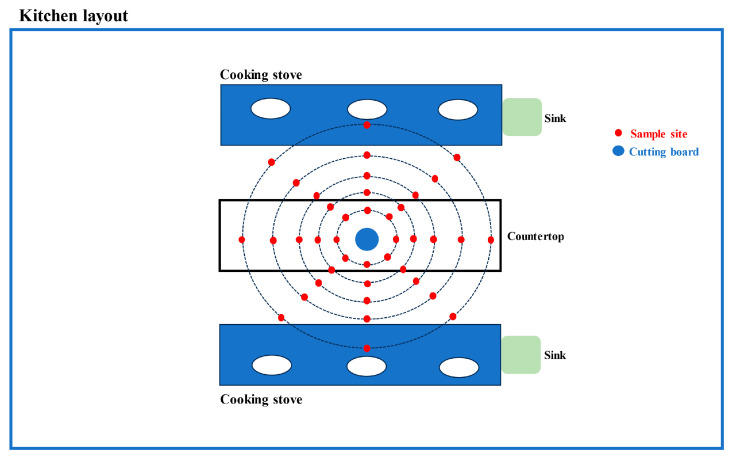
Schematic diagram of sampling points at monitoring locations during simulated fresh chicken processing in the model kitchen. Blue circles represent cutting boards, red circles represent monitoring points. The distance between each point is 30 cm, forming a cross shape with equidistantly spaced monitoring points.

**Figure 2 foods-12-03245-f002:**
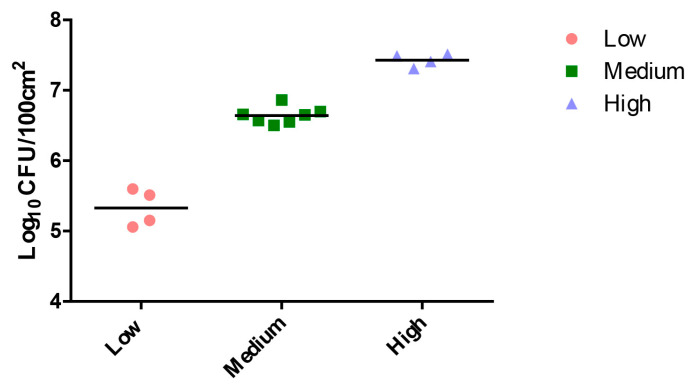
Concentration of *C. jejuni* in chicken carcasses (log_10_ CFU/100 cm^2^) initially contaminated with the same concentration of ACTCC 12,662 at low, medium, or high levels (log_10_CFU/mL), measured in 15 meat samples analyzed after treatment.

**Figure 3 foods-12-03245-f003:**
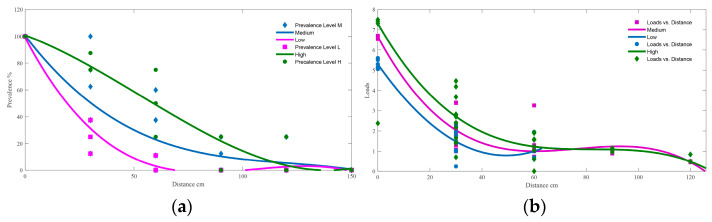
Detection rate (**a**) and Loads (**b**) of *C. jejuni* (log_10_CFU/100 cm^2^) in cotton samples compared to the level of *C. jejuni* (log_10_CFU/100 cm^2^) on the chicken carcasses for 40 sample sites in a model kitchen after cutting chicken.

**Figure 4 foods-12-03245-f004:**
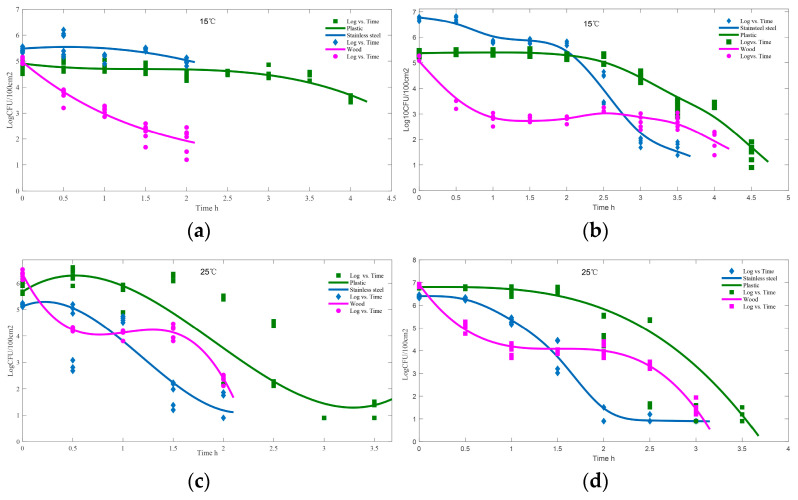
Survival of *Campylobacter* on boards made of different materials. The cultured standard strain ATCC12662 broth was diluted with PBS (**a**,**c**) and chicken juice (**b**,**d**), and then 500 μL of bacterial suspension was inoculated on the surface of different materials Finally, the materials were placed in an incubator at 15 °C (**a**,**b**) or 25 °C (**c**,**d**). Mean values are shown.

**Table 1 foods-12-03245-t001:** Socio-demographic characteristics of the survey participants %.

	Total Sample (*n* = 154)	Jiangsu Province (*n* = 36)	Guangdong Province (*n* = 48)	Fujian Province (*n* = 33)	Guizhou Province (*n* = 17)	Hunan Province (*n* = 20)
1. Gender						
Male	70.78	69.44	79.17	72.73	64.71	55.00
Female	29.22	30.56	20.83	27.27	35.29	45.00
2. Age						
18–25	89.61	75.00	97.92	93.94	82.35	95.00
26–35	5.84	16.67	0.00	6.06	0.00	5.00
36–45	1.95	2.78	0.00	0.00	11.76	0.00
46–60	2.60	5.56	2.08	0.00	5.88	0.00
3. Education						
Junior high school education	3.25	0.00	8.33	0.00	5.88	0.00
Senior high school education	3.90	8.33	2.08	0.00	11.76	0.00
Vocational education	48.05	27.78	79.17	57.58	17.65	20.00
College education	39.61	50.00	8.33	36.36	64.71	80.00
Graduate education	5.19	13.89	2.08	6.06	0.00	0.00

**Table 2 foods-12-03245-t002:** Answer frequencies of hygiene practices of chefs reported while preparing chicken.

Items	Total Sample (*n* = 154)	Jiangsu Province (*n* = 36)	Guangdong Province (*n* = 48)	Fujian Province (*n* = 33)	Guizhou Province (*n* = 17)	Hunan Province (*n* = 20)
Type of catering service						
Hotel	58.44	50.00	72.92	72.73	17.65	50.00
Restaurant	35.71	44.44	20.83	24.24	70.59	45.00
Fast food restaurant	5.84	5.56	6.25	3.03	11.76	5.00
Type of meat						
Fresh chicken	33.12	19.44	41.67	39.39	29.41	30.00
Frozen chicken	16.23	22.22	12.50	18.18	11.76	15.00
Chicken slaughter in Wet market	38.96	41.67	33.33	30.30	58.82	45.00
Chicken breast or leg	11.69	16.67	12.50	12.12	0.00	10.00
Whether the whole chicken had been eviscerated or not						
Yes	87.01	94.44	83.33	81.82	88.24	90.00
No	12.99	5.56	16.67	18.18	11.76	10.00
Before cutting, whether or not the chicken had been washed						
Yes	91.56	88.89	95.83	87.88	88.24	95.00
No	8.44	11.11	4.17	12.12	11.76	5.00
Type of cutting board for chicken preparation						
Plastic	50.00	41.67	62.50	48.48	41.18	45.00
Wood	44.16	55.56	33.33	45.45	47.06	45.00
Stainless steel	1.95	2.78	0.00	6.06	0.00	0.00
Other	3.90	0.00	4.17	0.00	11.76	10.00
Before chicken preparation, whether or not boards used for cutting other meat						
Yes	40.91	38.89	39.58	36.36	41.18	55.00
No	59.09	61.11	60.42	63.64	58.82	45.00
After other meat preparation, method of cleaning boards						
Washing with running tap water	33.12	33.33	45.83	24.24	11.76	35.00
Scrapping with knife	31.17	19.44	25.00	42.42	41.18	40.00
Washing with detergent	35.71	47.22	29.17	33.33	47.06	25.00
While preparing chicken, whether or not cloths used to clean cutting board						
Yes	48.05	61.11	41.67	45.45	41.18	50.00
No	51.95	38.89	58.33	54.55	58.82	50.00
Cleaning method for cloths						
Washing with running tap water	17.53	19.44	18.75	21.21	5.88	15.00
Washing with detergent	51.95	50.00	56.25	42.42	64.71	50.00
Boiling	29.22	30.56	20.83	36.36	29.41	35.00
No washing	1.30	0.00	4.17	0.00	0.00	0.00
After chicken preparation, method of cleaning boards						
Washing with running tap water	33.77	38.89	41.67	39.39	5.88	20.00
Scrapping with knife	24.03	19.44	16.67	18.18	41.18	45.00
Washing with detergent	41.56	41.67	39.58	42.42	52.94	35.00
No washing	0.65	0.00	2.08	0.00	0.00	0.00
After touching the chicken, (usual) method of cleaning hands						
No washing	3.90	8.33	6.25	0.00	0.00	0.00
Washing with running tap water	41.56	27.78	39.58	51.52	41.18	55.00
Washing with disinfectant	54.55	63.89	54.17	48.48	58.82	45.00
After chicken preparation, (usual) method of cleaning the knife						
Washing with running tap water	40.91	50.00	41.67	45.45	11.76	40.00
Washing with detergent	51.95	38.89	54.17	48.48	82.35	50.00
Clean with cloth	6.49	8.33	4.17	6.06	5.88	10.00
No washing	0.65	2.78	0.00	0.00	0.00	0.00

**Table 3 foods-12-03245-t003:** Different cleaning methods for reducing *Campylobacter* loads on cutting boards in 23 commercial kitchens in eastern China.

Cleaning Method	Stage	Total	Positive	Detection Rate %	Average Loads (Log_10_CFU/100 cm^2^)
cleaning with a knife (*n* = 17)	chicken	222	172	77.48	2.98 ± 0.93
	before cutting	17	4	23.53	2.45 ± 0.68 ^a^
	after cutting	17	16	94.12	2.97 ± 0.94 ^b^
	after cleaning process	17	9	52.94	2.10 ± 0.56 ^a^
cleaning with running tap water (*n* = 5)	chicken	67	58	86.57	3.25 ± 0.68
	before cutting	5	0	0	-
	after cutting	5	5	100	3.44 ± 0.85 ^a^
	after cleaning process	5	3	60	2.85 ± 0.47 ^a^
cleaning with disinfectant (*n* = 1)	chicken	12	12	100	4.03 ± 0.31
	before cutting	1	0	0	-
	after cutting	1	1	100	3.76 ± 0.00 ^a^
	After cleaning process	1	1	100	1.6 ± 0.00 ^a^

Different letters indicate significant differences among the groups (*p* < 0.05).

## Data Availability

The data presented in this study are available on request from the corresponding author.

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
