# Peer review of "Evaluation of Hygiene Practice for Reducing Campylobacter Contamination on Cutting Boards and Risks Associated with Chicken Handling in Kitchen Environment"

_foods, 2023, doi:10.3390/foods12173245_

Round 1

Reviewer 1 Report

The article submitted by Lai et al aimed to assess the handling practices of chefs across five provinces in China and identify the key factors contributing to the unsafe use of cutting boards, with particular emphasis on Campylobacter risk exposure. The research topic is relevant to the journal's scope and adds to the already known evidence on Campylobacter cross-contamination. Some remarks for improvement are proposed below.

-          Abstract: please shorten it and and structure it in one paragraph. Conclusion part in the abstract: “The practices employed by chefs increase the risk of pathogen contamination”; this is a subjective generalization – change to - based on the current study data, the practices employed by chefs play important role in Campylobacter transfer in the kitchen environment…… After reading the conclusion part, the reader will end up asking “and so??”; hence, you need to add a line of recommendation, maybe about strengthening hygiene culture, hygiene awareness,.. and so.

-          Keywords: risk area, risk time (they are rarely searched for terms – consider using risk exposure or other terms…

-          Line 46: remove undercooked (rare) and emphasize cross-contamination only.

-          Line 60: a few recent studies – does not read well, rewrite (some studies……)

-          Line 65: start with giving 3 lines about the situation of Campylobacter in humans in China (or prevalence data in chicken done recently in China).

-          Sentence in lines 68 to 70 is repeated in lines 75 to 77; please correct and adjust the whole paragraph to avoid repetition and add clarity to the aim of the work.

-          Material and methods: the first thing to read in this section should be “ethical approval”; the ethical approval permit number and the assigning authority should be clearly stated as the first thing in the material and methods section. Also, this is missing at the very end of the article!!

-          web-based survey – indicate the program/tool used to set it up; also, how the chefs were assigned, and their contact information was obtained is not clearly stated! “through an online approach???”. In addition to human ethics approval, biological safety approval should be provided concerning the work done in the experimental kitchen.

-          in five provinces of China… is there an issue with listing them at least once in the methodology sections? (we can read them in the table, but better to state them in the text as well)…

-          Line 94: avoid discussion in this section “which aligns with the gender distribution of individuals……….”

-          Table 1: the age grouping is not correct; 18-25 then 25 to 35 (25 then is in both groups, and this is not a correct way of presenting the results). The grouping should be 18-25; 26-35; 36-45; 46-60

-          Line 102: on what criteria the chefs were selected?

 In eastern China?? State which provinces (as not all readers are familiar with the Geography of China) – and why only in eastern China (if logistics then indicate, no problem).

-          Line 109: the protocol must be explained briefly here.

“samples were taken from the chicken carcasses, which had an approximate size of 500cm2”; what kind of sample? And is it not more meaningful to refer to carcass weight rather than size (which is meant to indicate surface area, not actual size!!)??

-          Line 112: “modified CCDA, Preston” These are two different agar media you are listing – please correct or clarify!!

-          Line 113: replace microaerophilic with microaerobic (also explain how it was generated)

-          Line 114: plated twice x (plated in duplicate)

-          using 225 mL of Buffered Peptone 115 Water – regardless of the cloth sample weight??? Then how could you compare the results? It is not logical (for the reader) to imagine a uniform clothes weight among such a number of chefs!!!

-          “was inoculated into 9 mL of Bolton broth supplemented with 5% defibrinated sheep blood”; why was this used here, and for the other samples, the diluent was PBS? What is the impact of such unnecessary variation in protocols?

-          “dilution range of 15-300 colony-forming units (CFU) per plate” – I am not sure this is needed (or even if it is correct). What if you found five colonies on 10^-1 and 0 on 10^-2???

-          “a minimum of five” – this kind of writing seems to be very idealistic and worries me (that means that on all retained plates, you counted more than 5 colonies!!!). It is better to be rational; replace “a minimum of five” with “up to five”.

-          Line 124: genes in italic

-          500-mL bag containing the Campylobacter ATCC12662 130 cultures; what concentration? What load per ml??

-          and stored at a temperature of 4°C – for how long??

-          L 137: Following the handling of the chicken??? By whom?? And what is/how was it such handling happening??

-          It should be noted that no significant differences were observed among the different supermarkets or the whole chickens purchased (data not shown)??? Was that because freezing is expected to kill Campylobacter quickly, regardless of country, setting, and supermarket!!

-          Line 168: “To compare the variances in the positive rate of Campylobacter detection” – this is very confusing, given that all of the protocols you explained were quantitative (count data), so why is your analysis now focused on positive/negative!!!

-          The statistics do not sound correct at all: The chi-square test cannot accommodate the analysis of counts, and basically, all your methodology is based on generating count data. So, what you describe here does not fit the research goal that the reader expects from this study.

-          The above points make me, as a reviewer, believe that the presentation of your results is wrong. For example, Line 228: “Fortunately, the average Campylobacter loads on the boards noticeably decreased from 2.97±0.94 Log10CFU/100cm2 to 2.10±0.56 229 Log10CFU/100cm2 (P<0.05). In the previous lines, you compare counts and justify significant differences with P-value based on what kind of test?? You stated in the statistics section that you used the Chi-square test (which is no sense at all). This analysis is wrong and not trusty as presented, and it is not acceptable. This is also raising speculations about the validity of your results (as the same mistake keeps on and on).   

Moderate editing of English language required

Author Response

Dear Editors and Reviewers:

Thank you for your letter and for the reviewer’ comments concerning our manuscript entitled “foods-2534930”. Those comments are all valuable and very helpful for revising and improving our paper, as well as the important guiding significance to our researches. We have studied comments carefully and have made correction which we hope meet with approval. Revised portion are marked in red in the paper. The main corrections in the paper and the responds to the reviewer’s comments are as following attachment

Reviewer 2 Report

I revised the manuscript. The method and method are written in the appropriate language and fluency. It just needs to be cited, I suggested it on the article. The results are summarized and tables and figures are sufficient. The results of the research were discussed by giving enough literature. In the article, the Introduction part is sufficient, I only suggested the necessary resources. References are written according to journal rules.

Author Response

Dear Editors and Reviewers:

Thank you for your letter and for the reviewer’ comments concerning our manuscript entitled “foods-2534930”. Those comments are all valuable and very helpful for revising and improving our paper, as well as the important guiding significance to our researches. We have studied comments carefully and have made correction which we hope meet with approval. Revised portion are marked in red in the paper. The main corrections in the paper and the responds to the reviewer’s comments are as following:

Point 1: The method and method are written in the appropriate language and fluency. It just needs to be cited, I suggested it on the article. The results are summarized and tables and figures are sufficient. The results of the research were discussed by giving enough literature. In the article, the Introduction part is sufficient, I only suggested the necessary resources. References are written according to journal rules

Response 1: Thanks for you good suggestion and careful reading. Line41, Line 61, Line 109, we have added the reference and information in the paper.

Reviewer 3 Report

The manuscript describes research conducted to determine the influence of cutting board composition on the spread of Campylobacter. The literature is rich with research linked to cross-contamination from cutting board surfaces.  Regardless, this research expands on the issue. 

The introduction and discussion must contain background on regulations in China concerning food preparation for direct consumption by the public.  In the United States the FDA Food Code considers practices acceptable for retail food establishments (restaurants).  Without context concerning current standards the manuscript fails to provide sufficient relevance.

Specific Comments:

Line 127. The justification for irradiation of the chicken prior to inclusion in the study is weak. The natural microflora of the chicken would be expected to influence interaction of Campylobacter.    Were the investigators hesitant to use non-treated chicken since it would be more difficult to identify inoculated verse naturally occurring Campylobacter?

Line 185.  What are the CFDA or local guidelines/regulations with respect to types of cutting board materials permitted.  Seems that wood is not permitted based on previous literature.

Line 220. What type of training do chefs undergo with respect to understanding basic cleaning and sanitizing practices for food contact surfaces.

Line 225.  In the manuscript cleaning refers to removal of gross physical debris.  It would appear that no or few chefs actually sanitized the cutting boards prior and after use.

Line 366.  Sanitizers used at concentrations know to be effective for treatment on food contact surfaces have not never resulted in development of resistant strains.

Generally acceptable.

Author Response

Dear Editors and Reviewers:

Thank you for your letter and for the reviewer’ comments concerning our manuscript entitled “foods-2534930”. Those comments are all valuable and very helpful for revising and improving our paper, as well as the important guiding significance to our researches. We have studied comments carefully and have made correction which we hope meet with approval. Revised portion are marked in red in the paper. The main corrections in the paper and the responds to the reviewer’s comments are as following attachment.

Round 2

Reviewer 3 Report

No additional comments.

 Please be certain that revisions are reviewed for language. Note in the example below "Contamination" the "C" should be lower case.

"surfaces showed Campylobacter spp. Contamination in commercial"